# Diagnosis of the Pneumatic Wheel Condition Based on Vibration Analysis of the Sprung Mass in the Vehicle Self-Diagnostics System

**DOI:** 10.3390/s23042326

**Published:** 2023-02-20

**Authors:** Krzysztof Prażnowski, Jarosław Mamala, Adam Deptuła, Anna M. Deptuła, Andrzej Bieniek

**Affiliations:** 1Faculty of Mechanical Engineering, Opole University of Technology, 45-758 Opole, Poland; 2Faculty of Production Engineering and Logistic, Opole University of Technology, 45-758 Opole, Poland

**Keywords:** vibroacoustics, diagnostics, data classification, decision trees, pneumatic wheel logic tree

## Abstract

This paper presents a method for the multi-criteria classification of data in terms of identifying pneumatic wheel imbalance on the basis of vehicle body vibrations in normal operation conditions. The paper uses an expert system based on search graphs that apply source features of objects and distances from points in the space of classified objects (the metric used). Rules generated for data obtained from tests performed under stationary and road conditions using a chassis dynamometer were used to develop the expert system. The recorded linear acceleration signals of the vehicle body were analyzed in the frequency domain for which the power spectral density was determined. The power field values for selected harmonics of the spectrum consistent with the angular velocity of the wheel were adopted for further analysis. In the developed expert system, the Kamada–Kawai model was used to arrange the nodes of the decision tree graph. Based on the developed database containing learning and testing data for each vehicle speed and wheel balance condition, the probability of the wheel imbalance condition was determined. As a result of the analysis, it was determined that the highest probability of identifying wheel imbalance equal to almost 100% was obtained in the vehicle speed range of 50 km/h to 70 km/h. This is known as the pre-resonance range in relation to the eigenfrequency of the wheel vibrations. As the vehicle speed increases, the accuracy of the data classification for identifying wheel imbalance in relation to the learning data decreases to 50% for the speed of 90 km/h.

## 1. Introduction

An important issue in the scientific field of operation of technical objects is the evaluation of their technical condition without disassembly [1]. For this purpose, work processes and phenomena accompanying the operation of such a technical object are most often used for research [2,3]. This type of approach to monitoring their condition is particularly used during the operation of various types of vehicles as well as in machinery and equipment [4,5]. The article [6] presents a Bayesian-optimized discriminant analysis model for classifying and monitoring tool status in three user-defined classes. Hyperparameter matching is activated by a Bayer optimization search. The authors demonstrated the feasibility of using Bayesian optimization algorithms to fine-tune the classification model, making it industry-ready.

In order to avoid operating a vehicle in poor technical condition, its operation should be monitored in real time during operation. For this purpose, vehicles are equipped with an OBD self-diagnostic system. The essence of the system is to quickly diagnose defects that have a negative impact on the environment [7], which are mainly related to the operation of the vehicle’s drive unit. The on-board diagnostics system does not diagnose many other factors affecting the vehicle’s technical condition, which would affect driving comfort and safety. The pneumatic wheel is one such component, which, due to its design, has not yet been supplemented with a diagnostic system. Enabling monitoring of its operation in the OBD system, this would make it possible to identify its technical condition in terms of imbalance. The pneumatic wheel is an essential component of many systems used in a vehicle, such as the chassis, steering, braking and suspension systems.

Long-term operation of a vehicle with a wheel in poor technical condition, characterized by imbalance, shape errors or inhomogeneity in the circumferential stiffness of the tire, can lead to damage to other components of the chassis, as shown in Figure 1. Therefore, it is important to perform an early identification of the occurrence of a wheel malfunction during its normal operation based on the first signals indicating its failure. An important factor in the process of inferring the technical condition of a device on the basis of certain symptoms is the data classification algorithm and inference system.

Keeping in mind the continuous development of the road infrastructure and the known concepts related to electric vehicle drive systems (e.g., the mounting of electric motors in the wheels of a car [8,9]), require the pneumatic wheel to maintain a relatively low mass. This is because the pneumatic wheel, together with other components such as brake discs, calipers, bearings, pins and electric motors are part of the vehicle’s unsprung mass. For this reason, reduction in the weight of the entire wheel should be pursued because in the event of a malfunction of the wheel, its operation in an unbalanced condition may have serious consequences for the remaining structural components of the vehicle. One main consequence can be accelerated wear and damage to other components. This article presents a procedure for identifying the malfunction of a pneumatic wheel of a road vehicle caused by its imbalance under conditions of its normal operation using inductive knowledge acquisition (machine learning). The identification of wheel malfunction was based on the vibration of the vehicle’s body using a three-axis acceleration sensor and an optical speed measurement head. The recorded acceleration signal was subjected to spectral analysis using power spectral density and selected rotational frequency ranges of the pneumatic wheel. Own machine learning algorithm based on decision trees within the car’s self-diagnostic system was used for inference. Stationary tests were carried out using a chassis dynamometer, as well as road validation tests taking into account the disturbances resulting from road surface irregularities. The variation of amplitude values at successive harmonics of the fundamental frequency was indicated and used as learning data in the formulated algorithm. A regression model of the diagnostic parameter as a function of driving speed was developed, validated and statistically verified.

## 2. Related Works and Other Methods

One of the many examples of extending the diagnostic functions of on-board diagnostic systems is the use of the pneumatic wheel pressure change signal as a parameter for identifying its technical condition to monitor the safety of moving a vehicle [2]. In this way, pneumatic wheel diagnostics uses the method of detecting the change in voltage of a piezo resistive pressure sensor, which provides a greater accuracy due to built-in temperature compensation, filtering and self-calibration and identifies the pressure drop inside the tire. An indirect method is to monitor the change in the rolling radius of the pneumatic wheel; the deviation of the value of this radius from the average value is identified as a pressure drop in the tire. In the article [10], the authors presented an advanced method for identifying tire pressure using a smart tire with three-axis accelerometers. The relationship between tire pressure and the characteristic frequency of a rolling wheel was established. As a result of testing at different tire pressures, the authors found that the value of the dominant frequency for unsprung mass increases with a change in tire pressure. The article [11] presented a different method of multidimensional identification of signal characteristics in the analysis of vibration properties of the vehicle floor panel as a sprung mass, which allows selection and separation of signal components in multiple domains. The presented method makes it possible to define the signal depending on the characteristics of stationariness and non-stationariness as well as precise time localization of sprung frequencies. Having access to such results, it is possible to analyze the vibration of the body as a result of excitation by road bumps. In the article [12], the authors proposed a new method for estimating vehicle dynamic parameters and longitudinal tire stiffness. The developed model of the longitudinal dynamics of the vehicle uses the acceleration/deceleration of the sprung mass of the body to estimate the position of its center of gravity as well as the longitudinal stiffness of the tire. Taking into account the continuous control of the vehicle, the developed vehicle model can be used to estimate the tire stiffness in turnings, the vehicle’s moment of inertia about the yaw axis and the moment of inertia about the longitudinal axis. The two-step estimation method proposed in this article makes it possible to simultaneously obtain all the parameters needed for modeling vehicle dynamics.

One solution for detecting faults in wheels, tires and related suspension components in vehicles is the use of a smartphone and the analysis of data in the frequency domain. The authors of [13] demonstrated that the developed classification tree model built using the data of Fourier features achieved 79% classification accuracy on the test data for the assumed vehicle speed of 96.56 km/h.

Many studies use mass vibrations and advanced mathematical methods of signal analysis to identify the condition of many vehicle components. The authors of [14] presented the potential related to using vibration signal parameters in assessing the clearance of elements in the car suspension system. Time spectrum analysis was proposed in order to determine the frequency band related to the natural vibrations of the car body. The paper describes diagnostic models that allow the assessment of shock absorber mounting to the car body. The paper [15] presents the diagnosis of valve clearances of an internal combustion engine based on vibration signals using machine learning algorithms. The vibration signals were obtained from three-axis vibration acceleration transducers located on the engine head. The obtained time waveforms of the vibration signals were parameterized for an engine operating under different loads, speeds and valve clearances. A similar approach using enhanced variation mode decomposition (VMD) and a bispectral algorithm for vibration signal analysis was described in [16]. The analysis of the vibration signal in the frequency domain (e.g., determining the failure frequency of characteristic rolling bearing components [17]) is an effective way to detect bearing malfunctions without disassembly. In the study reported in [18], the authors presented the use of decision trees for rolling element bearing diagnostics based on selected vibration signal characteristics. The paper [2] presents a unified methodology for incremental learning of new information from evolving databases. The proposed methodology was presented on data sets from an automotive electronic throttle control subsystem. The authors evaluated the performance of adaptive classification techniques when the new data have the same error classes and the same features and when the new data have new error classes but with the same set of observed features. The issue of identifying the tram wheel shape error was analyzed in [15]. The authors presented an algorithm according to which it is possible to detect a flat wheel during the passage of a streetcar. The proposed method is based on the measurement and analysis of the vibration signal in the time and frequency domain. Hilbert transformation was used in the algorithm. Experimental studies and analysis of the results of the method showed high efficiency in detecting wheel flattening and ovalization errors. In [19], a new diagnostic algorithm was proposed to evaluate the condition of the bearings of the drive shaft of a road vehicle during operation. The diagnostic parameter was selected based on vibration measurements and the results of machine learning analysis. The upper and lower limits of the diagnostic parameter were determined, and based on this, the vehicle user would receive information about the upcoming wear and tear and total failure of the bearing. The study in [20] describes a method for detecting shock absorber damage in the primary suspension of a rail vehicle based on the analysis of vertical vibrations of the bogie. The experimental data obtained were used to determine the normal operating area and damage area of the shock absorber in the primary suspension of the bogie. The article [21] describes the analysis of serious damage to truck tires. The authors propose the use of noninvasive diagnosis methods and driver assistant system to monitor the technical condition of tires.

Currently, most research related to data classification for inference focuses on data mining algorithms [22,23,24,25] and knowledge discovery from data. Knowledge discovery is interdisciplinary in nature and uses results from statistics, machine learning, data visualization, databases and processing of uncertain information (e.g., fuzzy set theory or rough set theory) [26]. An effective process of creating and extracting this knowledge requires the selection of an appropriate expert. Currently, the interdisciplinary nature of these activities is evident not only in the area of combining research techniques but also in the implementation encountered in the engineering practice. One such example is the purposeful selection by an expert responsible for the innovation of a given solution. In the case of the construction of expert systems, this aspect should be taken into account when choosing the person responsible for its design, and an appropriate expert should be selected [27]. Machine learning and statistics are of great importance nowadays due to the provision of algorithms for knowledge discovery [28]. On the other hand, the specificity of knowledge discovery from databases provides new research problems for both machine learning and databases. In practice, it turns out that the explicit acquisition of an extensive knowledge base is a difficult and time-consuming task [29,30]. In contrast, observations and descriptions of examples of specific decision-making situations are generally much easier to access and are more objective.

## 3. Identification of the Pneumatic Wheel Condition

### 3.1. Research Methodology

A vehicle moving on the road is subjected to a constantly changing environment, which generates vibrations of unsprung and sprung masses of a random (stochastic) nature. Thus, statistically describable kinematic excitation is created, the value of which (at certain moments) can reach significant values perceptible to the driver. In a situation where these values are felt by the driver and are a consequence of overcoming an obstacle, for example, in the road surface, it results directly from the reaction of the vehicle’s chassis or suspension system. In the paper [31], the authors used the vibration of the sprung mass of the vehicle to determine the condition of the road surface, and their own methodology based on the measurement of longitudinal accelerations and a method of classifying the condition of the road surface was developed. However, in a situation where a vehicle is driven on a road that is in good condition and vibration of the unsprung or sprung mass is experienced, it is too late for diagnostic assessment because the defect has already taken place and the vibration amplitude levels have been significantly exceeded. Therefore, this article presents a two-stage testing approach. In the first stage, bench testing was carried out using a MAHA MSR500 dynamometer to eliminate the interference from roadway irregularities. The test station was equipped with a single measuring roller with a diameter of 350 mm. The linear speed of the test vehicle was determined by the speed of the dynamometer’s brake roller. During the measurement, the tire front of the pneumatic wheel of the tested axle was placed on the dynamometer roller. This allowed the determination of values for the acceleration of the unsprung and sprung masses (of the body) in the vertical direction as a result of excitations both from the rolling balanced wheel and a wheel with a preset imbalance. Measurements of vehicle body vibrations were carried out for selected speeds in the range from 50 km/h to 100 km/h. During testing, a constant speed was maintained for a period of 10 s. The tests were carried out both for a balanced wheel and using a wheel with a preset imbalance on only one drive wheel in the front axle in order to determine the levels of diagnostic estimators for a well-functioning and malfunctioning pneumatic wheel. In the second stage, tests were carried out on a different road surface, and its effect on the value of the amplitude of the acceleration of the body of the test vehicle was determined. The methodology was based on measuring and analyzing the component signals from the acceleration sensor (*x*, *y* and *z*) as a database and using them for machine learning algorithms. Importantly, road verification tests of wheel imbalance assessment based on vibrations of the sprung mass of the vehicle body were performed on a road with a good road surface with no obvious cavities or transverse or longitudinal ruts.

### 3.2. Measurement System

The vehicle applied during the tests had a MacPherson-type front column suspension with a single wishbone and a telescopic coil spring shock absorber. A measurement application developed in LabView software was used for the experimental tests. It allows the recording of the signal from the three-axis acceleration sensor (*x*, *y* and *z* axes) and the linear speed of the vehicle (using an optical measurement head). The schematic of the system for measurement and data acquisition is shown in Figure 2. It consists of a 3DM-GX3-25 sensor for acceleration measurement (Table 1), whose measurement range is ±50 m/s^2^ with a resolution of 0.05 m/s^2^, and a head for non-contact measurements of the speed of the body relative to the road L-350 AQUA (Table 2) connected to the NI SCC-68 input module connected to the NI USB 6212 data acquisition device.

The integrated sensor for measuring acceleration by the direct method (3DM-GX3-25) used in the study was made with MEMS technology and is insensitive to the influence of external and internal interference, e.g., from sensor signal conditioning systems due to the use of a sensor unit with PWM pulse output. It has a built-in processor that provides a static and dynamic orientation of its measurement axes through a measurement synthesis algorithm.

### 3.3. The Main Concept of Experimental Measurements

The main research objective was to measure the vibration of the sprung mass of the vehicle body based on the assumptions of the active–passive experiment. In the presented experiment, a correlation was made between the values of the harmonic components of the amplitude spectrum at a preset imbalance and the current speed of the vehicle. The evaluation of the technical condition of the tested pneumatic wheels was carried out on machines used for static and dynamic determination of the balance with an accuracy of 0.005 kg. This method is currently used in the case of repairing or changing tires. In this case, it is only possible to detect the degree of wheel imbalance, which can be corrected. Measurements were carried out for two test cases. In the first one, the vehicle was equipped with properly balanced pneumatic wheels. In this case, the technical condition of the vehicle could be described as fully operational. Next, measurements were carried out on the vibration of the sprung mass of the body with a preset wheel imbalance.

The structure of the analysis of the recorded data is shown in Figure 3. The power spectral density was determined for the recorded signal from the acceleration sensor. Strong periodic components are visible in the resulting spectrum. It was also found that the instantaneous speed of the vehicle and the periodic components in the frequency spectrum were consistent with the angular wheel velocity. The fundamental frequency was determined by the parameter f_0_. Also visible are the successive harmonics of the fundamental frequency, which were determined as f_1_, f_2_, f_3_ and f_4_. The occurrence of successive harmonics indicates the presence of resonance caused by the rolling wheels of the vehicle. It is also important to select the speed range of the vehicle due to the resonant frequency of the pneumatic wheel. For the purposes of this analysis, the authors adopted selected speed ranges between 50 km/h and 100 km/h. The inference algorithm was based on a comparative analysis of the learning data contained in the database and the input data. The database was a compilation of features of the signal subjected to analysis in the frequency domain at a known preset imbalance weight. The study was conducted to determine for which vehicle speeds the expert inference method would be most beneficial.

## 4. Analysis of the Body Vibration Signal

The power spectral density was determined for the recorded components of the vehicle body acceleration. The analysis of the obtained spectrum showed that the peak value of the amplitude of the fundamental harmonic (f_0_) and subsequent harmonics (f_1_, f_2_, f_3_ and f_4_) depended on the degree of preset wheel imbalance and the linear speed of the vehicle. Finally, the spectrum of the signal varied in terms of peak values and its slenderness (shape), which is related to the energy contained in the analyzed signal. The stochastic excitation that occurs due to the unevenness of the road caused an increase in the signal power for individual harmonics with respect to the frequency of wheel rotation.

For speeds of 50 km/h, the peak PSD values of the spectrum for all analyzed axes (*x*, *y* and *z*) in the band up to 70 Hz did not exceed 0.12 (Figure 4). An increase in amplitude in the frequency range of wheel rotation f_0_ was visible in the *y*- and *z*-axis directions for a given imbalance of 0.08 kg mass. Also visible were the amplitudes of the third harmonic (f_3_) of the *y*-axis for a balanced wheel and for a preset imbalance of 0.04 kg.

For a speed of 60 km/h, there was an apparent increase in the amplitude value at a preset mass imbalance of 0.04 kg and 0.08 kg at the first harmonic (f_1_) in the *x*- and *z*-directions (Figure 5). The occurring increase in the peak value of the spectrum in harmonics f_1_, f_2_, f_3_ and f_4_ was related to the circumferential stiffness of the wheel. The preset imbalance with simultaneous tire inhomogeneity can cause a significant increase in the amplitudes of the considered PSD spectrum ranges. An increase in the peak value of the PSD can be seen on the fundamental harmonic f_0_ in the *y*-direction, which is related to the occurrence of wheel imbalance.

A significant increase in the peak values of the PSD spectrum in the *y*- and *z*-direction can be seen at 70 km/h (Figure 6). There was then a resonance of the unsprung mass (the pneumatic wheel), which manifested itself as an increase in the PSD peak value at f_0_ to 0.23 for the *y*-axis and 0.19 for the *z*-axis. An increase in the value of the f_1_ amplitude in the *z*-axis direction for the balanced wheel indicates the occurrence of inhomogeneous tire stiffness.

The subsequent increase in speed to 80 km/h, 90 km/h and 100 km/h showed a significant dominance of amplitudes in the *y*- and *z*-directions. In the longitudinal direction (*x*-axis), PSD peak values still did not exceed 0.15 (Figure 7, Figure 8 and Figure 9).

In the second stage of the research for the adopted test scheme of signal analysis of the components of the vibration signals of the sprung mass of the vehicle body in the *y* and *z* axes, the power value of the spectrum of the analyzed signal was determined for the adopted harmonic ranges (Figure 10). The frequency range window of the analyzed spectrum was ±1.5 Hz with respect to the determined fundamental harmonic and its successive multiples.

The obtained values of the power field of the PSD signal with respect to the considered ranges of the fundamental frequency f_0_ and its subsequent harmonics (f_1_, f_2_, f_3_ and f_4_) represent the distribution of the signal power field at a certain state of wheel imbalance. For the longitudinal axis (*x*-axis), the values of the power field do not significantly depend on the speed of the vehicle. The assumed state of the wheel (balanced and with a preset imbalance) does not change the power field values for individual harmonics. This was particularly evident for speeds of 70 km/h, where the distribution of PSD values did not depend on the state of the wheel (Figure 11).

For the *y*-axis, the dependence of the PSD value on the condition of the wheel and the speed of the vehicle was equal to 60 km/h, 80 km/h and 90 km/h (Figure 11). An increase in the preset imbalance results in an increase in the PSD value of the signal of the fundamental harmonic f_0_ and the subsequent f_1_, f_2_ and f_3_. An increase in the preset imbalance results in an increase in the signal power field.

An analogous increase in PSD values with a change in speed was seen for the *z*-axis (Figure 12). This was particularly evident at speeds of 80 km/h, 90 km/h and 100 km/h.

The proposed solution makes it possible to compare selected ranges of adopted harmonics of the analyzed signal spectrum. The obtained results for verification road tests are shown in Figure 8 (*y*-axis) and Figure 9 (*z*-axis). The obtained data were then subjected to inference analysis using inductive decision trees and an expert system.

## 5. Application of Decision Rules to Classify Data

Decision rules are induced from data sets representing information about a set of objects called learning examples, which are described by a set of attributes. The problem considered in terms of machine learning is to find rules that determine whether an object belongs to a certain subset called a (decision) class or concept. Decision rules induced from examples are logical expressions represented in the form:*if* certain conditions are met, *then* a certain decision is made

If knowledge of a field is limited and one does not have a known classification of objects, it is possible to use data analysis methods, such as cluster analysis, to find classes of similar objects [32]. The most important part of this process is associated with the data mining stage (Figure 13) and the use of a proper inductive algorithm [33] that can offer relationships and patterns in the prepared dataset.

The decision tree algorithm is used in machine learning to extract knowledge from examples. The task of decision trees is to identify the correct voltage control areas (VCAs) based on the numerous branching power distribution variants in the power system. Classification involves finding a way to map the data into a set of predefined classes of power distribution variants. Based on the contents of the database, a model (classification tree) is built, which is used to quickly identify the correct VCAs.

One of the most popular algorithms ID3 [34] (and its later version C4.5 [35]) was based on information theory. This algorithm recursively divides the space of examples into subspaces, representing the division in the form of an optimal decision tree. It is necessary to specify the objects (phenomena, problems, etc.) to be classified, the attributes that describe the objects, and the values that each attribute will take. The choice of the attribute used to divide the analyzed set is made on the basis of the value of the measure called information gain (1),
(1)$$I(S_{L})=-\sum_{i=1}^{\left|S_{L}\right|}\frac{\left|S_{Li}\right|}{\left|S_{L}\right|}\cdot \log_{2}\left(\frac{\left|S_{Li}\right|}{\left|S_{L}\right|}\right),$$
where $\Delta I(S_{L},a)=I(S_{L})$ is calculated from the entropy of information *I(S_L_*), where *S_L_* is a collection of learning cases. |*S_Li_*| is the number of cases describing the *i-th* decision class.

A decision was made to train the tree with help of entropy. Although there was also the possibility of using a modified classifier algorithm. The inner workings of the accuracy of the Gini algorithm are similar to the workings of entropy in a decision tree. In the decision tree algorithm, both are used to build the tree by splitting according to the relevant features, but there is quite a big difference in the calculation of the two methods. The Gini count of the features after splitting can be calculated using this formula. In addition, the Gini count is calculated based on the classic CART algorithm, which is very easy to estimate. In this case, the entropy method was chosen to obtain a simple tree when choosing the test. In addition, when determining entropy, the following can be obtained:-Functions measuring the difference between the set E (of examples) and the sets into which this set is divided according to the value of the attribute being evaluated due to the frequency distributions of the class decisions;-Functions measuring the difference between different subjects of the set E (formed by the value of the evaluated attribute) due to the frequency distributions of class decisions;-Functions measuring the statistical independence between the distribution of class decisions and the division of E into subsets.-Other reasons for choosing entropy include the following aspects:-It favors splits with fewer observations but many unique values;-In its calculation, the probability of a class is weighted by logarithm of the base two of the probability of that class.

Entropy *I(S_L_, a)* for attribute *a* is expressed by Equation (2):(2)$$I(S_{L},a)=\sum_{m=1,\ldots ,S_{L}^{\left(m\right)}\neq 0}^{\left|S_{L}\right|}\frac{\left|S_{L}^{\left(m\right)}\right|}{\left|S_{L}\right|}\cdot I\left(S_{L}^{\left(m\right)}\right),$$
*|S_L_^(m)^|* is the number of instances of value *m* for attribute *a.*

In the approach proposed in this paper, a derivative of the C4.5 type decision tree building system [36] was used to determine the new feature from the original case description. The resolving function in two-dimensional space is a straight line running between clusters of points representing cases of different classes. In the case of two-dimensional space, the analyzed objects are described by two features. In the case of multidimensional space, the resolution essentially comes down to determining on which side of the decision boundary the unknown object lies. This matter is determined by the sign of the scalar product *s* (Equation (3)) of the weight vector *w* as well as by the image vector *x* and the cosine of the angle between them (Figure 14).
(3)$$s=\bar{w}\cdot \bar{x}=\left|\bar{w}\right|\cdot \left|\bar{x}\right|\cdot \cos\alpha ,$$

For more than two dimensions, in the formula is the sum of:(4)$$s=\sum_{i=1}^{d}w_{i}x_{i},$$
$$s=w_{1}x_{1}+w_{2}x_{2}+w_{3}x_{3}+\ldots +w_{d}x_{d}$$

The following hyperparameters of the decision tree were selected:-Max_depth (i.e., blocking the maximum depth of the tree)—set to one so that the main factor is the expansion of the tree height;-Min_samples_split—using the decision tree generation option, the minimum number of examples forming the leaf of the tree was assumed to be five, and the option to trim the tree by 25% was selected (this is a hyperparameter of DecisionTreeClassifier);-Min_samples_leaf—parameter responsible for the minimum leaf size after splitting. It is very similar to the min_samples_split hyperparameter; however, it is concerned with the minimum number of observations after splitting (this is a hyperparameter of DecisionTreeClassifier).-With DecisionTreeClassifier, the following were analyzed:-Max_depth—from 1 to 12 (10 values);-Min_samples_split—from 4 to 22 (18 values);-Min_samples_leaf—from 1 to 30 (30 values);-Criterion—entropy, Gini (2 values).

Finally, a total of 10·18·30·2 (i.e., 10,800) models were analyzed. Each of these models was trained. Some were trained for a shorter time (e.g., a small max_depth value), and some were trained for a longer time (e.g., max_depth value). Finally, among all available models, the maximum model was selected.

If we do not specify any value for the max_depth parameter until it reaches a number of observations equal to the min_samples_split parameter at a given node, the tree will stratify until the final leaves contain homogeneous subsets with min_samples_split equal to one. This is very unfavorable, especially for large sets composed of many variables, as the detailing of the model increases significantly. As the complexity of the tree increases, the model adapts more and more to the observations contained in the training set, and both the min_samples_split and min_samples_leaf parameters will affect the pruning of the tree to a similar extent as the max_depth.

Although the minimum percentage of harvest used for partitioning at a node vs. model accuracy when specifying the minimum parameter percentage of the set used for splitting at each node was set to a value >30%, the accuracy of the model would not be improved.

That is, by changing the value of the min_samples_split parameter to 0.3, a better result can be obtained. Note that the parameters are to some extent “interchangeable”, and we can use both max_depth and min_samples_leaf parameters to trim the decision tree.

In this study, the kNN algorithm was used in the final stage of inference for the expert system. The kNN algorithm involves the calculation of the distances in *d*-dimensional space between the classified object and all objects in the learning set, sorting the calculated distances in ascending order and then selection of the *k* smallest distances for the next stage of the algorithm. It was assumed that *k* must be an unlabeled odd integer. In cases where objects located in the vicinity of the analyzed object belonged to different categories, in order to determine the membership of the analyzed case in a particular class, a voting procedure was used.

## 6. Application of the Concept of Multi-Category Object Classification to the Diagnostics of the Pneumatic Wheel

In the procedure applied for developing an expert system for wheel diagnostics, the assumptions presented in [27] for selecting an expert designer of this system were used. For this purpose, three potential experts were studied, and the most suitable one was selected.

### 6.1. Generation of Induction Trees and Rules

In the first stage, an algorithm based on entropy growth was used. A learning file and a testing file were created. The learning file concerned road tests for a balanced wheel and one with a preset imbalanced mass of 0.04 kg and 0.08 kg. An example of the learning data record is shown in Table 3. Input attributes (in) are *x*, *y* and *z* values. Output attributes (out) are information about the condition of the wheel. Output attributes refer to the values of the preset imbalance of 0.04 and 0.08 kg and the correct balance. In Table 3, the data refers to a wheel imbalanced by a mass of 0.04 kg.

In the next step, rules were created for the induction tree. Figure 15 shows an example of the rules applied for the decision tree for the *z*-axis. The fragment of the inductive decision tree shown refers to the learning file. The tree hierarchically (Figure 15) defines the most important parameter, which is classified at the very top of the induction tree. In our case, it was *z*. Then, starting with the most important parameter, a classification into further sub-attributes occurs in relation to the most important data values. Thus, it is worth noting that if the database (i.e., our measurement data) is different, the trees will inductively determine a new rank of importance, and then it may turn out that the parameter *z* is not the most important. In the above example, the division of further attributes after the parameter *z* occurred relative to the value *z* = 0.313, and further classification division for the parameter *y* occurred relative to the value *y* = 0.057. Finally, already at the last stage of classification, the classification division for *y* occurred relative to the value *y* = 0.114.

In the next step of the inference analysis, the process of generating a test file was carried out. The learning file concerned the values of the determined signal power of selected harmonics obtained under stationary and road conditions for a balanced wheel and with a preset imbalance for each axis (*x*, *y* and *z*).

### 6.2. Creating an Expert File

The expert file consists of a previously generated learning file and a previously generated testing file for selected speed ranges. The most important input components are the rules previously generated for the learning file and for the testing file (learning rules and testing rules). The output data (out) are the values of imbalance 40 and 80, a value attributed to wheel balance and the probability value for the occurrence of a given imbalance or balance of a pneumatic wheel. The data fragment in the expert system is shown in Figure 16.

Then the appropriate parameters were selected for expert system file generation. It was assumed that for the expert system, the size of the learning file would be 90%, while the size of the testing file would be 10%. In the learning and testing dataset, it is necessary to determine whether the measured value of the parameter is within the norm. A typical split is 90–10% or 70–30%. When using a split of 100–0%, there is no testing file, and the program does not work. Using a 50–50% split, the program cannot distinguish which file is learning and which is testing. The use of minimal differences when dividing the data (90–10%) does not adversely affect the result of the inference.

For the analyzed cases, category membership was assigned based on the closest objects to the classification. If out of the five closest objects, three belonged to class **A** and two belonged to class **B**, (where A and B stand for balancing or unbalancing of the wheel with the corresponding value), class **A** was assigned to the analyzed object. The issue of assigning an unknown (analyzed) case to the correct category is determined by probability. Figure 17 shows an example of classification in terms of multi-category objects. If a value of *k* = 3 was obtained, then two values indicated that the analyzed result (categorized attribute value) belonged to class ■, and only one value indicated that it belonged to class **⨯**. Then such a received result is categorized into class ■. If, on the other hand, a value of *k* = 5 was obtained, then three values indicated that the analyzed result (categorized attribute value) belonged to class **⨯**, and two values indicated that it belonged to class ■. In this case, the analyzed object is assigned to class **⨯**.

The analyzed input data represented by the determined power values for each component axis: *x*, *y* and *z* are described by the *XYZ* vector:(5)$$\vec{XYZ}=\left\{ \begin{array}{l} \bar{X}=(x_{1},x_{2},x_{3},\ldots ,x_{h})\\ \bar{Y}=(y_{1},y_{2},y_{3},\ldots ,y_{h})\\ \bar{Z}=(z_{1},z_{2},z_{3},\ldots ,z_{h}) \end{array}\right.,$$

In the next step of the calculations, the distance between metrics, representing the distance between each pair of elements of this data set, is determined. It is important to determine the shortest possible distances connecting these data. The developed structure uses the Minkowski metric:(6)$$h={\left[\sum_{i=1}^{n}{\left(x_{i}-z_{i}\right)}^{p}\right]}^{\frac{1}{p}},$$

On this basis, the probability of the occurrence of a given value can be taken into account. In the classification of multi-category objects, it is also necessary to carry out optimization of selected parameters that can significantly affect the recognition of the class of the unknown object, namely the number of neighbors, the voting method and the distance measure method. For the graphs in the developed expert system, the Kamada–Kawai [17] learning model was used. Nodes depict rules, and then the probability of setting a given value is determined. The unordered rule-based correlation on the dependency graph is shown in Figure 18. In addition, the assumption was made that in the space under study, all edges should have as few intersecting edges as possible by assigning forces to a set of edges and a set of nodes based on their relative position and then using these properties to cover a set of classes. Figure 19 shows the ordered rule-based dependency according to the Kamada–Kawai model.

In the next step of data analysis, the probability value was determined. The target operation of the system was to determine the probability of the occurrence of a given phenomenon (the probability of balancing or unbalancing of the wheel with the corresponding value of this probability) on the basis of the entered values of power data for each axis (*x*, *y* and *z*).

As a result of the program, the presented induction tree determined that the most important component affecting the classification of the final decision is the *x* component, which for the generated knowledge base is located in the root of the decision tree.

By analyzing the recorded data for each axis of acceleration (*x*, *y* and *z*) (Figure 20), information about the correspondence with the data contained in the database was obtained. An example of component analysis for a speed of 50 km/h is shown in Figure 21. The classification analysis carried out in the first step showed that the analysis of the *y*-axis component had a lower value of classification performance (0.057) than for the *x*- and *z*-axes (a value greater than 0.057). In the next step, the system made an assessment for the other components (*x*- and *z*-axes). As a result, a higher data classification correctness occurred for the *x*-axis (a value greater than 0.06). As a result of the procedure, the system showed that the greatest conformity of the analyzed input data relates to the components of the *x*-axis.

In the developed expert system with an array architecture, the source of data in the array architecture is the knowledge base. In the developed program, it is possible to use any number of knowledge bases contained in the form of a table as the results of measurements for the *x*-, *y*- and *z*-axes. The system determines the occurrence of imbalance with a mass of 0.04 kg or 0.08 kg, the state of balance and the accuracy of its evaluation. As a result of data classification, information about the probability of occurrence of a wheel imbalance condition based on recorded vehicle body vibrations was obtained. Figure 22 shows an example of the classification of the expert system as the level of accuracy in terms of percentage of identification of input data against the data contained in the master database for selected ranges of vehicle speed.

For a speed of 50 km/h, the classification accuracy of the input data against the learning files contained in the database was 100% in most cases. This result was obtained both for a balanced wheel and one with a preset imbalance. For the speed of 70 km/h, a significant decrease in the accuracy level to values of 50% and 25% was evident. This is a result of the resonant frequency of the pneumatic wheel, which occurs at speeds around 70 km/h. In the range of higher speeds (from 90 km/h to 100 km/h), a variation in the level of accuracy of condition identification was evident. Indications of both 25% and 50% can be seen. The obtained indication of the wheel imbalance condition based on the recorded database for the vehicle could complement the vehicle’s self-diagnostic system under normal operating conditions. If an identification accuracy level of more than 75% is obtained, the system informs the driver via an indicator light on the dashboard that the vehicle’s wheel condition needs to be checked.

The method applied in this paper to estimate the conditional prediction error of a classifier was the so-called split sampling method. It consists of splitting the available data once, according to a predetermined proportion, into a learning and test sample. A classifier is constructed based on the learning sample. It is then used to predict the membership of objects from the test sample. The prediction error is expressed as the proportion of misclassified object from the test sample x_0_ = (t_0_, y_0_) in the total size of this sample. The advantage of this method is its simplicity and the fact that it does not involve extensive calculations, as the classifier is constructed only once. Its disadvantage, however, is that each object is assigned to only one of the analyzed samples. This has a significant impact on the results obtained, especially if there are outliers in the collected data, as is the case with vibration testing. However, this method can only be used if the data set is sufficiently wide to allow the separation of a sufficiently large number of independent sets for teaching and testing [37].

For example, in [6], the authors used discriminant analysis for TCM (tool condition monitoring). A novel approach in the paper was to use a Bayes-optimized discriminant analysis model for classification and monitoring.

Dynamic data provide an easy way to access a family of sample data. Such families in an analysis can be primitive data types or, as in our case, data arranged in sequential arrays. Processor layout and data distribution are important for performance-oriented parallel computing, but high-level language support to help programmers solve these problems is often insufficient. Therefore, papers should consider various high-level language constructs—grids (trees), distributions and areas—that allow programmers to manipulate the processor layout and data distribution in their calculations. For example, in [38], the authors presented grids as an abstraction of process sets, regions as an abstraction of index sets and distributions as an abstraction of the mapping from index sets to process sets. Each of these is a first-class concept, supporting dynamic reallocation and redistribution of data and dynamic manipulation of the processor set. The paper also considered the application of these constructs to the solution of several motivating parallel programming problems. In [39], the authors acquired spindle vibrations corresponding to correct and incorrect (with defects) cutter configurations in real time. These data were transformed to the time-frequency domain and further processed by the proposed architectures in the graphical form, i.e., spectrogram. The model was trained, tested and validated considering different data sets. In particular, the authors proposed CNN architectures, which were trained as spectrogram images. In this way, a classification model was developed. The paper showed that classification based on CNN architectures can be applied to ideal cutting tools. In [40], the authors presented an extension of the decision tree model, considering six different states of a single-point cutting tool in terms of tool vibration. In particular, they considered hyper and mini parameters. Larger ‘min’ parameters and smaller ‘max’ parameters supported the regularization of the model making it generalizable. The results achieved indicate that the augmentation methodology, i.e., tuning the hyperparameters, was capable and suitable for classifying valid tools and made the model robust to unfavorable data domains.

In contrast, in [41], the authors designed an event-driven algorithm in a VBE environment. They then applied decision tree generation using the J48 algorithm. In a practical approach, different tool conditions were categorized using different algorithms based on ‘supervised trees’.

## 7. Conclusions

This paper presented a method for identifying the occurrence of a tire wheel imbalance based on vehicle body vibrations using an expert program. For this purpose, the signal power field values were determined for selected ranges of determined harmonics in the power spectral density spectrum with respect to the angular velocity of the wheel. Then an induction system was used to generate induction trees. In this way, learning examples were generated. Induction trees classified the input data as power values for individual axes (*x*, *y* and *z*) as determined harmonic components (dependent on vehicle speed) from the point of view of the data at the output of the balanced wheel and with a preset imbalance (0.04 kg and 0.08 kg). From the induction trees, rules were created, and a formal description of the knowledge of a specific domain was obtained. At the same time, the analysis was applied to tests performed under normal vehicle operation conditions (on the road) and under stationary conditions (on a chassis dynamometer).

On the basis of these results, it was found that the highest probability of correct classification of the data against the adopted benchmark occurred for the vehicle’s pre-resonance speed of 50 km/h. The highest level of classification accuracy (with 100% efficiency) was obtained for all adopted states of the pneumatic wheel (balanced and with a preset imbalance of 0.04 kg or 0.08 kg). As the speed of the vehicle increased to a speed above resonance, the probability of correct indication varied from 50 to 75%. This is probably due to the greater deviation of components at higher speeds. The occurring phenomenon of resonance leads to a significant differentiation of spectrum components, which translates to the level of data classification accuracy in the developed reasoning system. The accuracy level dropped in this case to 50%. The least favorable vehicle speed for identifying wheel imbalance was above 70 km/h, where identification was realized with 25% efficiency.

Data analysis for super resonant velocities showed an average accuracy of 37% for 90 km/h and 30% for 100 km/h. Only the individual adopted data (1, 2, 6 and 7 for 90 km/h) showed a level of accuracy ranging from 50% to 70% up to 90%. This indicates the need to improve the classification process in the developed algorithm, especially taking into account higher vehicle speeds.

When data from a smaller number of harmonics was used (from f_0_ to f_3_) in the proposed system, the phenomenon of so-called data incompleteness may occur. As a result, the probability of correctly classifying the balance state of a pneumatic wheel is reduced. The problem of knowledge discovery in various types of databases, also known as deep data mining, is addressed in many different ways and by numerous artificial intelligence researchers. In the case of classification, perhaps knowledge discovery in data involves finding such features that best distinguish different classes from each other. The decision tree generation algorithm proposed in this paper makes it possible to generate sets of alternative trees (not just a single tree) to provide several different descriptions of the data in such a way that the expert system (or expert) can ultimately decide which set of rules best describes the problem under study.

As part of further research on the model of reasoning about the change in the condition of the pneumatic wheel, the authors plan to take into account other factors, such as the condition of the brake discs and changes in tire stiffness. Additional areas of model refinement will focus on a more detailed assessment of individual wheels as well as an attempt to quantify the severity of imbalances so that they can be repaired before permanent damage to other vehicle components ensues. In the future, the authors also plan to build an extensive expert system in Python and in an online environment, which will allow the concept of ongoing, continuous monitoring of vibroacoustic parameters to be realized. It should be noted that all information was collected during the measurement of data referring to a specific place of performing the measurements was compiled, an online preview of the values of the currently measured parameters was created and the measurement results were archived. This allows for further in-depth analysis. 

Additionally, in the future, the authors plan to use a wavelet transformation to reduce the noise created during the recording of acoustic emission signals accompanying the vibration test. It is planned to use a soft estimator as well as a hard estimator. In our opinion, the hard estimator will provide a much better representation of the measured signal (without attenuation) while leaving the signal slightly noisier. For example, the Donoho–Johnson method [42,43] can be used for such a purpose. This, in turn, may ultimately reduce the proportion of noise.

## Figures and Tables

**Figure 1 sensors-23-02326-f001:**
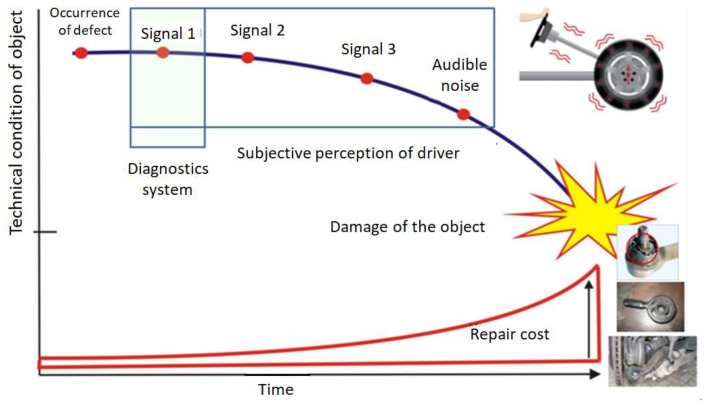
Timeline of early diagnostic decision of wear and tear.

**Figure 2 sensors-23-02326-f002:**
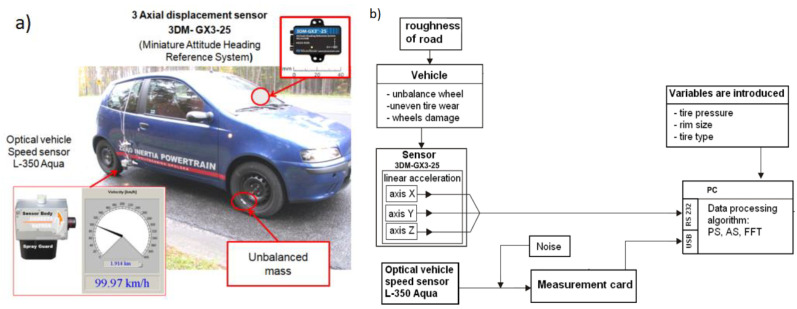
Test vehicle (**a**) and block diagram of the measurement system (**b**).

**Figure 3 sensors-23-02326-f003:**
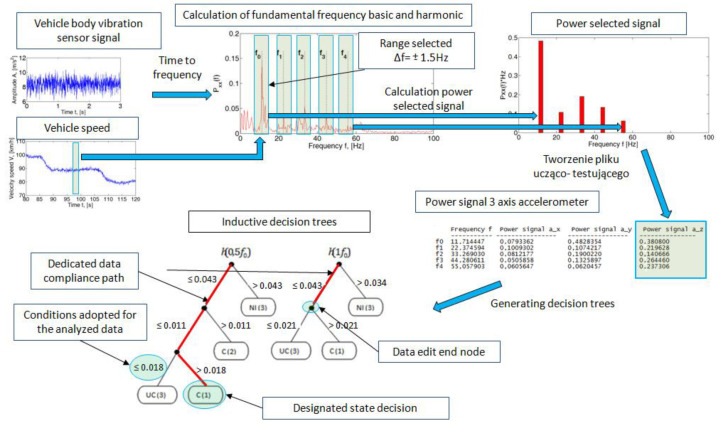
Data analysis structure.

**Figure 4 sensors-23-02326-f004:**
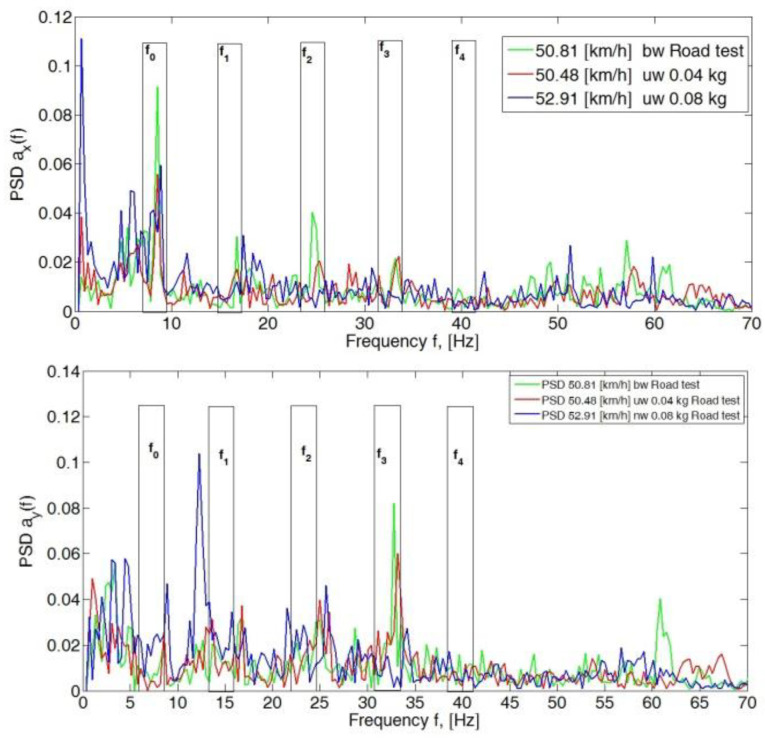
Power spectral density graph of the body acceleration signal for a speed of 50 km/h.

**Figure 5 sensors-23-02326-f005:**
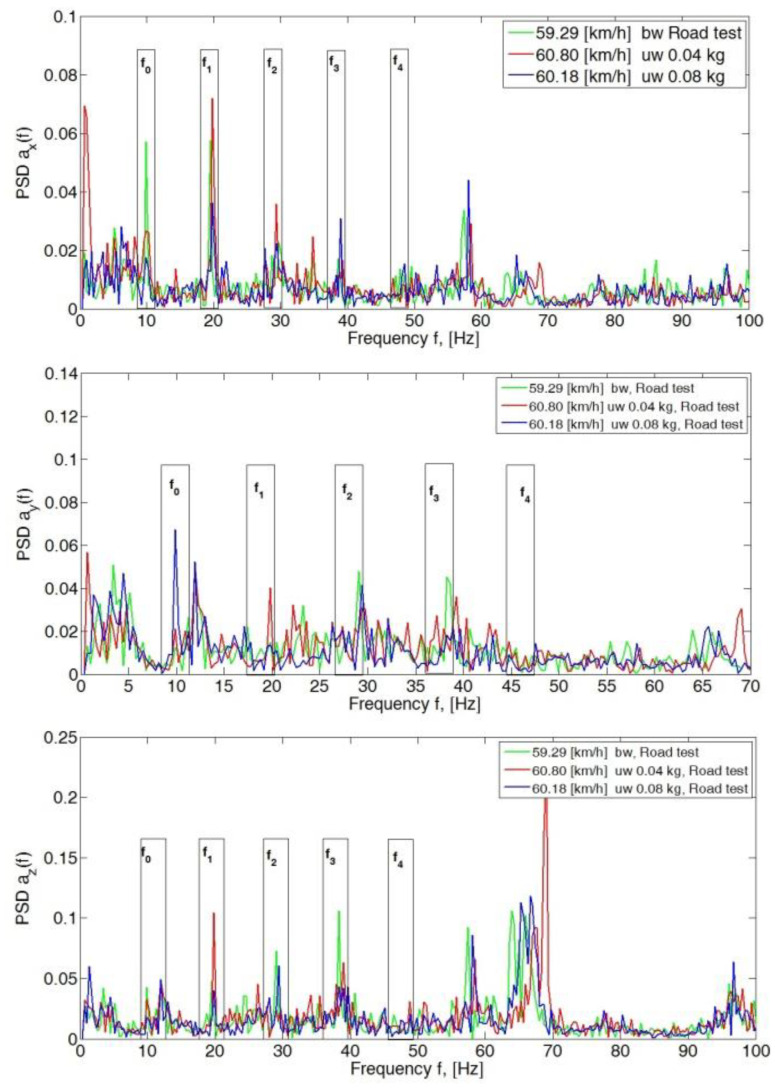
Power spectral density graph of the body acceleration signal for a speed of 60 km/h.

**Figure 6 sensors-23-02326-f006:**
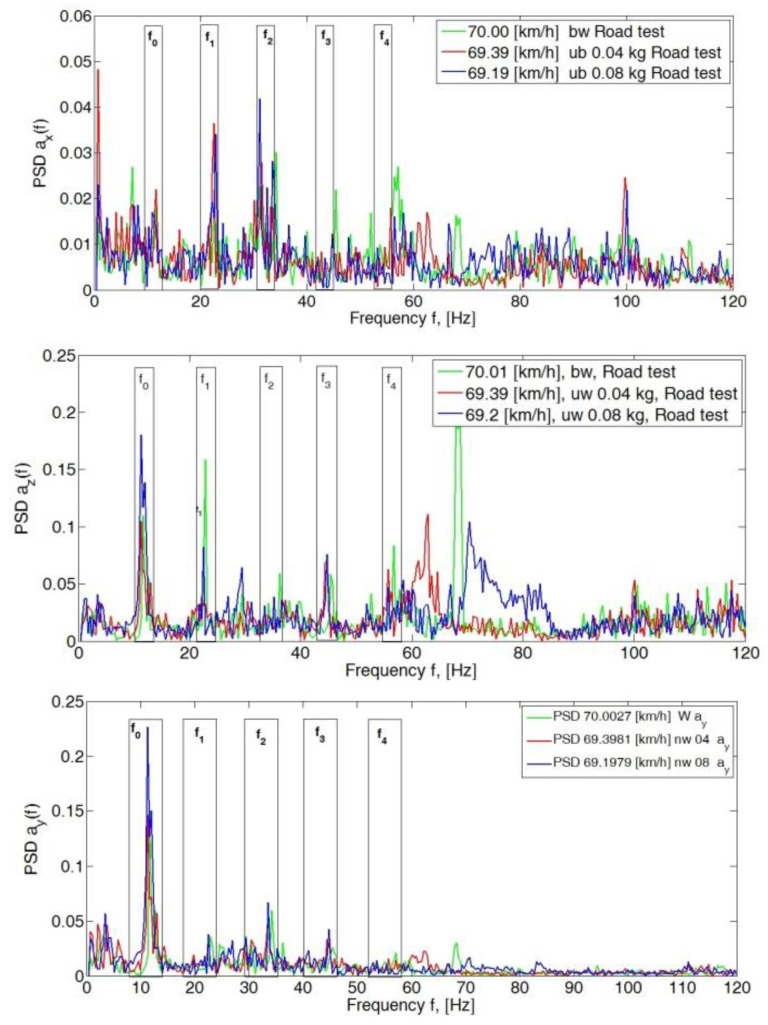
Power spectral density graph of the body acceleration signal for a speed of 70 km/h.

**Figure 7 sensors-23-02326-f007:**
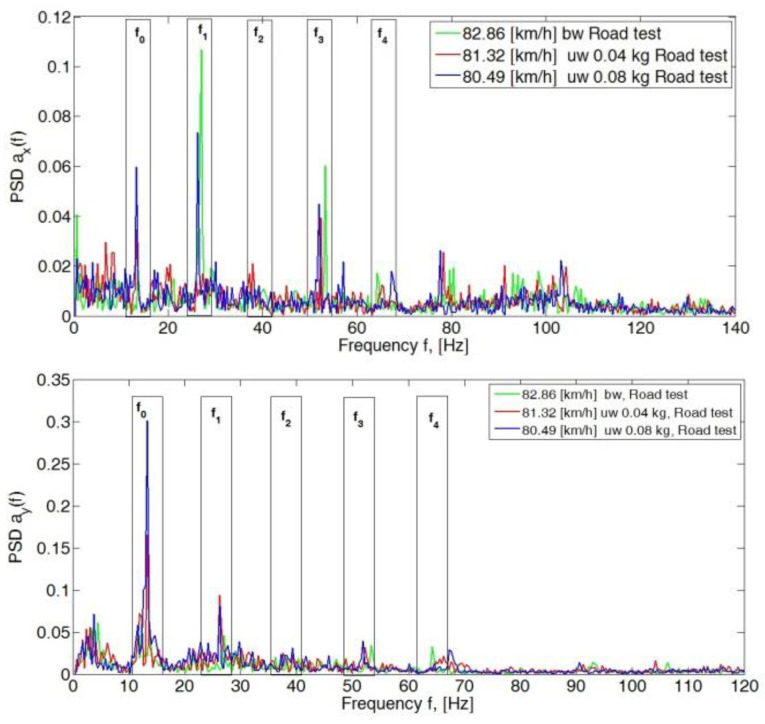
Power spectral density graph of the body acceleration signal for a speed of 80 km/h.

**Figure 8 sensors-23-02326-f008:**
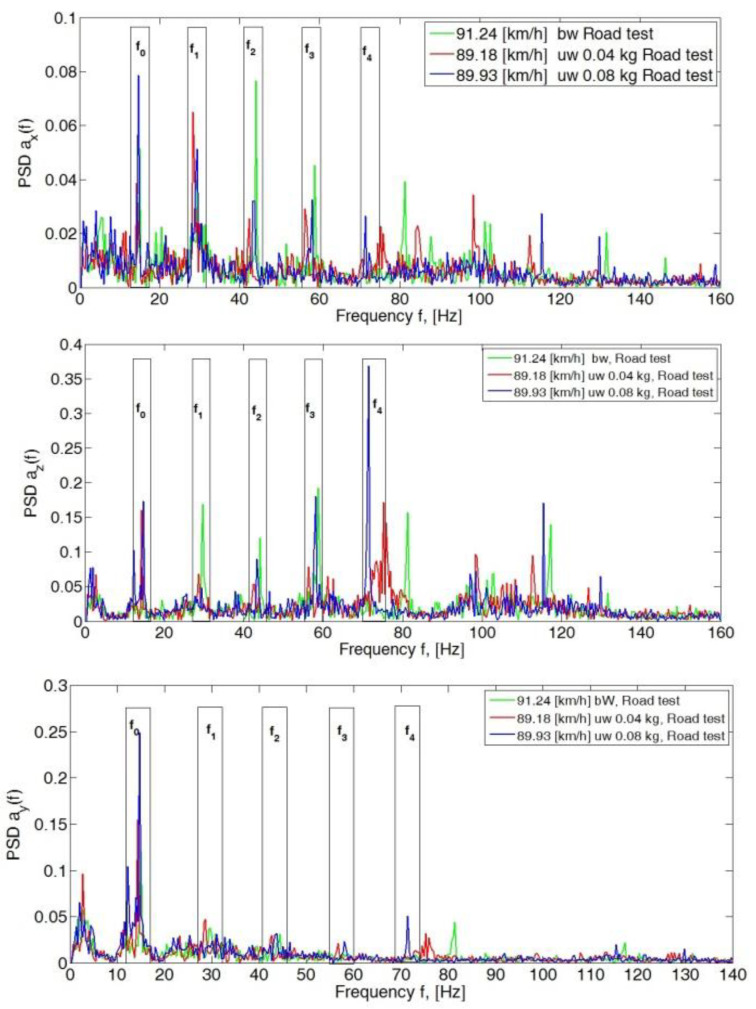
Power spectral density graph of the body acceleration signal for a speed of 90 km/h.

**Figure 9 sensors-23-02326-f009:**
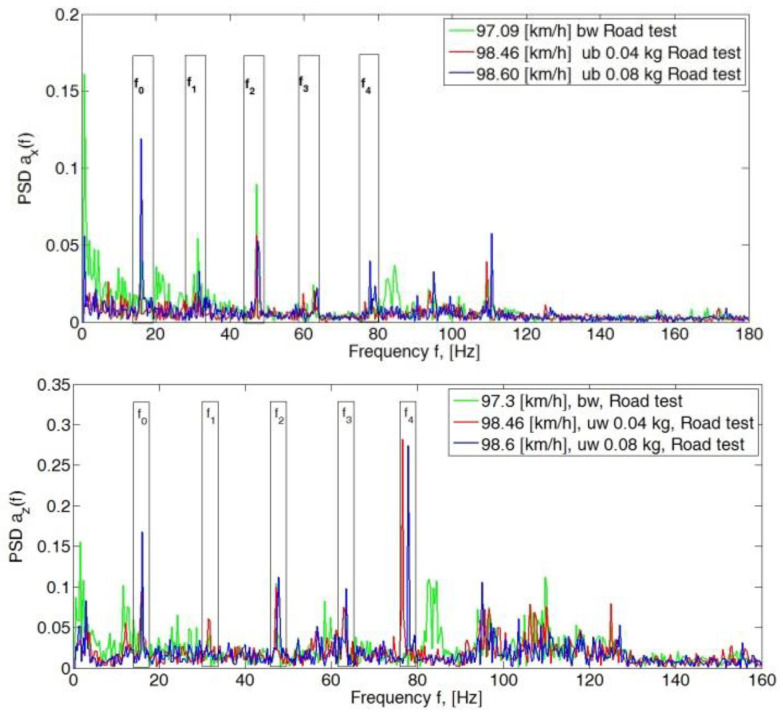
Power spectral density graph of the body acceleration signal for a speed of 100 km/h.

**Figure 10 sensors-23-02326-f010:**
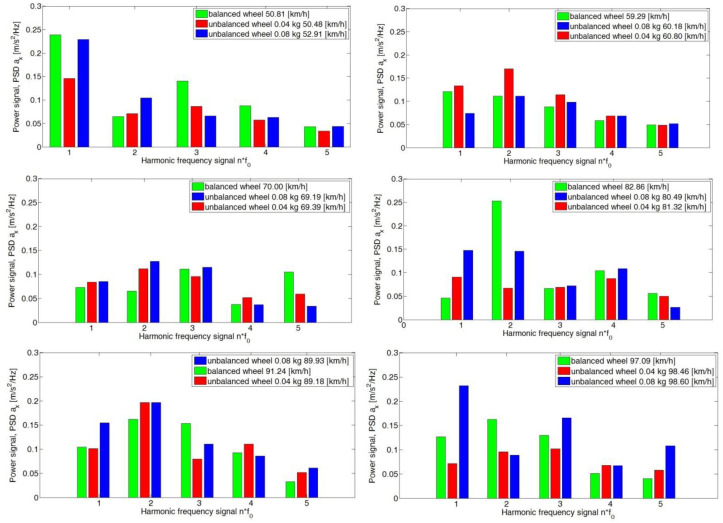
Signal power values of the analyzed harmonics of the *x*-axis for the adopted speed ranges for the road test.

**Figure 11 sensors-23-02326-f011:**
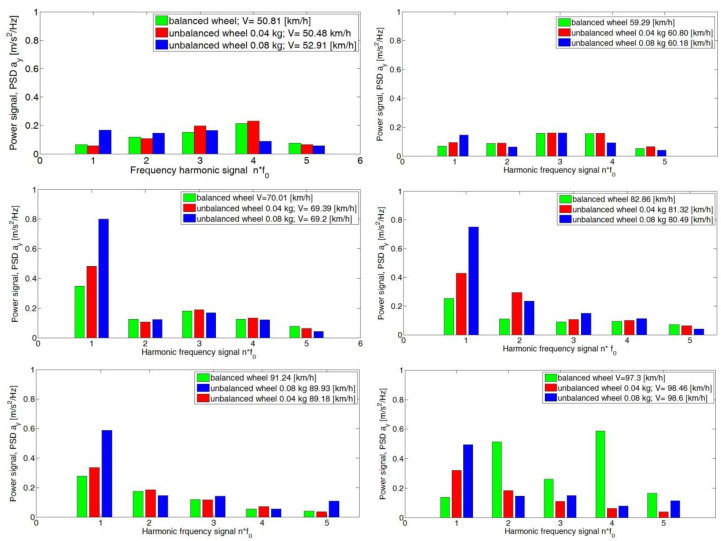
Signal power values of the analyzed harmonics of the *y*-axis for the adopted speed ranges for the road test.

**Figure 12 sensors-23-02326-f012:**
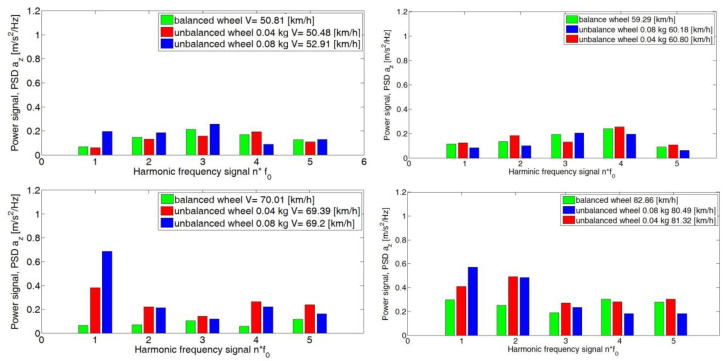
Signal power values of the analyzed *z*-axis harmonics for the adopted speed ranges for the road test.

**Figure 13 sensors-23-02326-f013:**
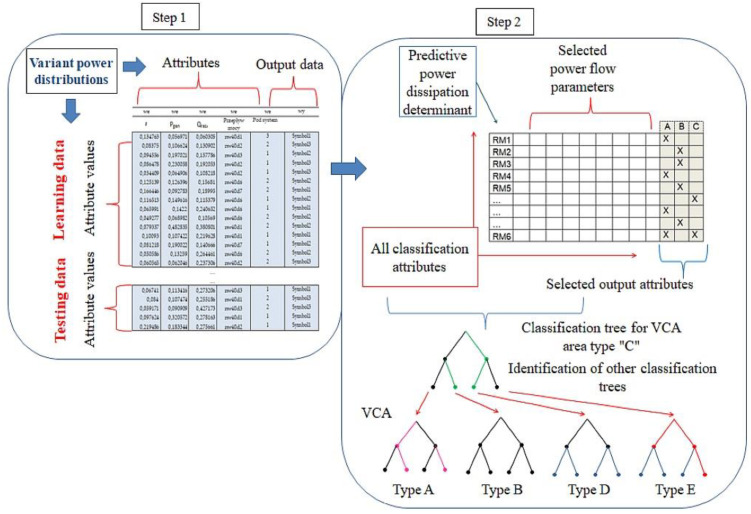
Knowledge generation in terms of decision tree classification in two stages.

**Figure 14 sensors-23-02326-f014:**
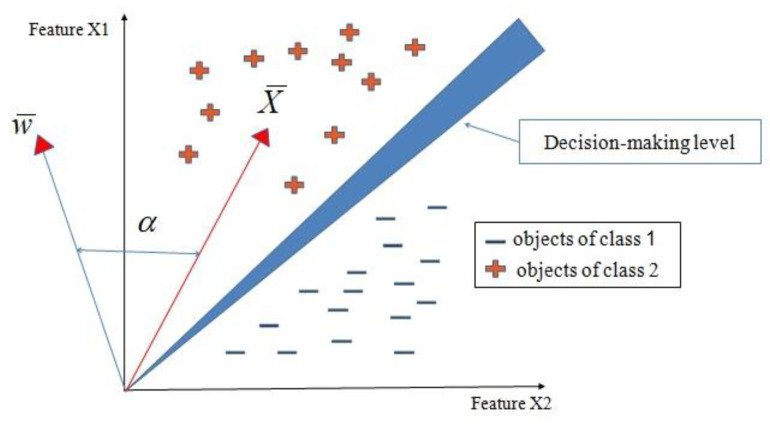
Example of classification for sample features.

**Figure 15 sensors-23-02326-f015:**
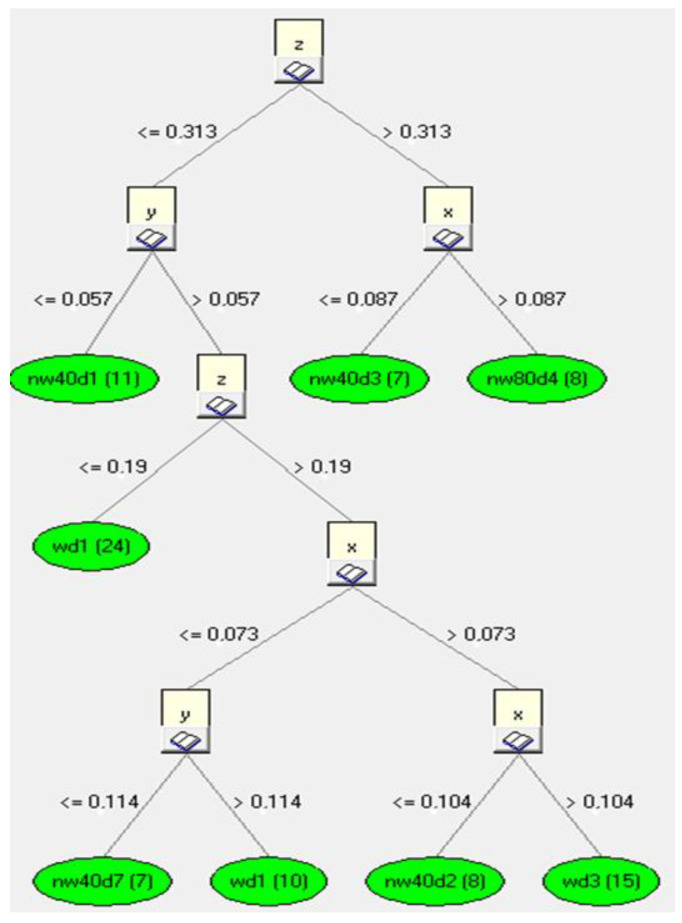
Induction tree of the analyzed *z*-axis data.

**Figure 16 sensors-23-02326-f016:**
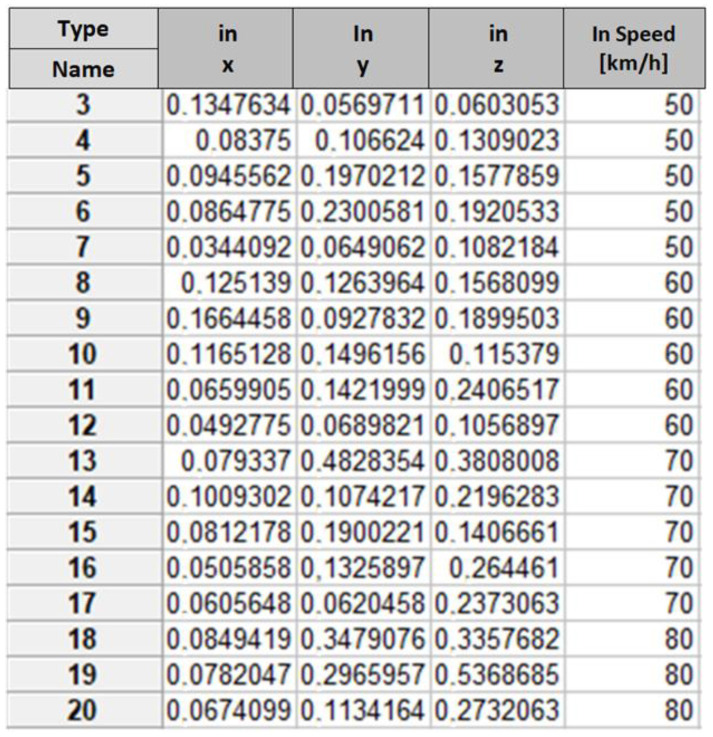
View of the data fragment in the expert system—input data (in).

**Figure 17 sensors-23-02326-f017:**
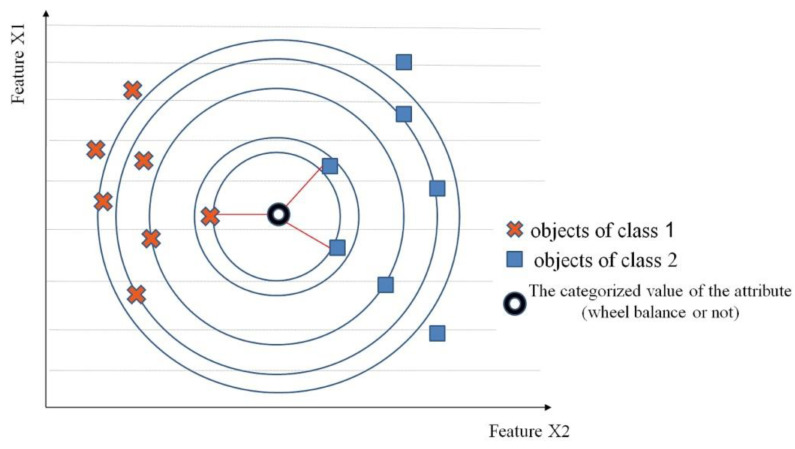
Application of multi-category object classification for sample classes.

**Figure 18 sensors-23-02326-f018:**
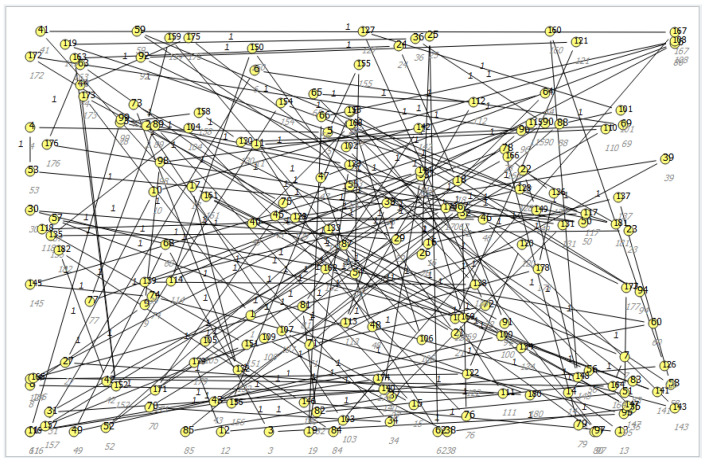
Search for rule-based dependency in an expert system.

**Figure 19 sensors-23-02326-f019:**
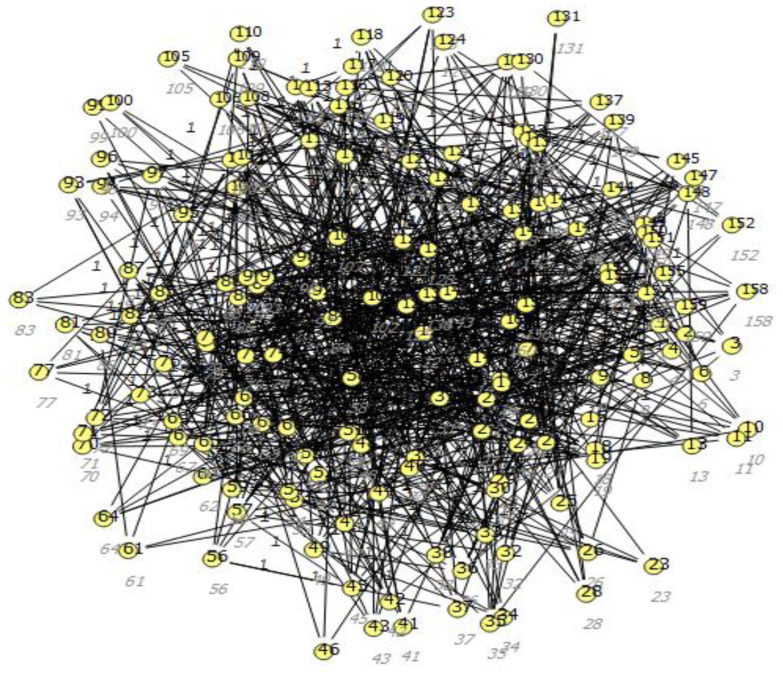
Ordered rule-based dependency according to the Kamada–Kawai model.

**Figure 20 sensors-23-02326-f020:**
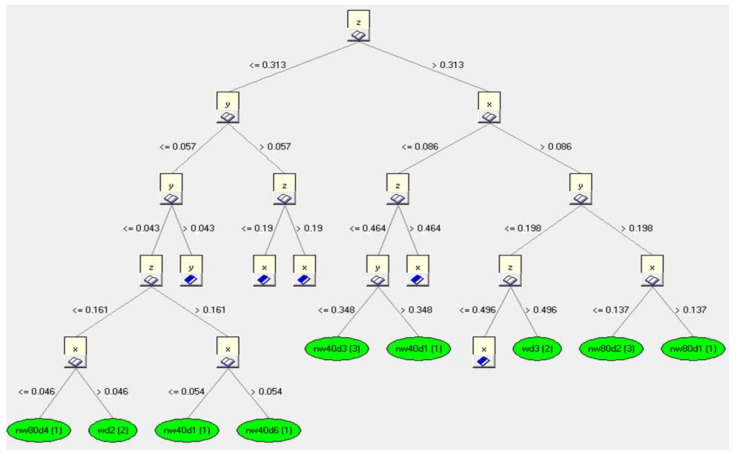
Data classification for *x*, *y* and *z* components at 50 km/h.

**Figure 21 sensors-23-02326-f021:**
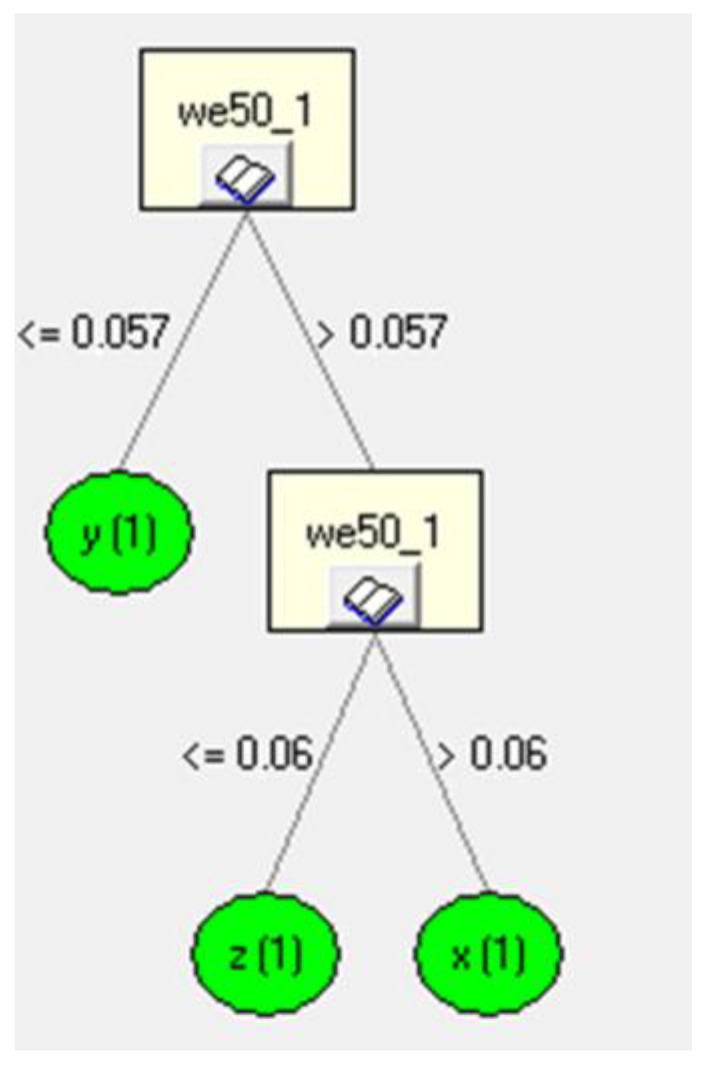
Analysis of the correctness of data classification against the database for the speed of 50 km/h.

**Figure 22 sensors-23-02326-f022:**
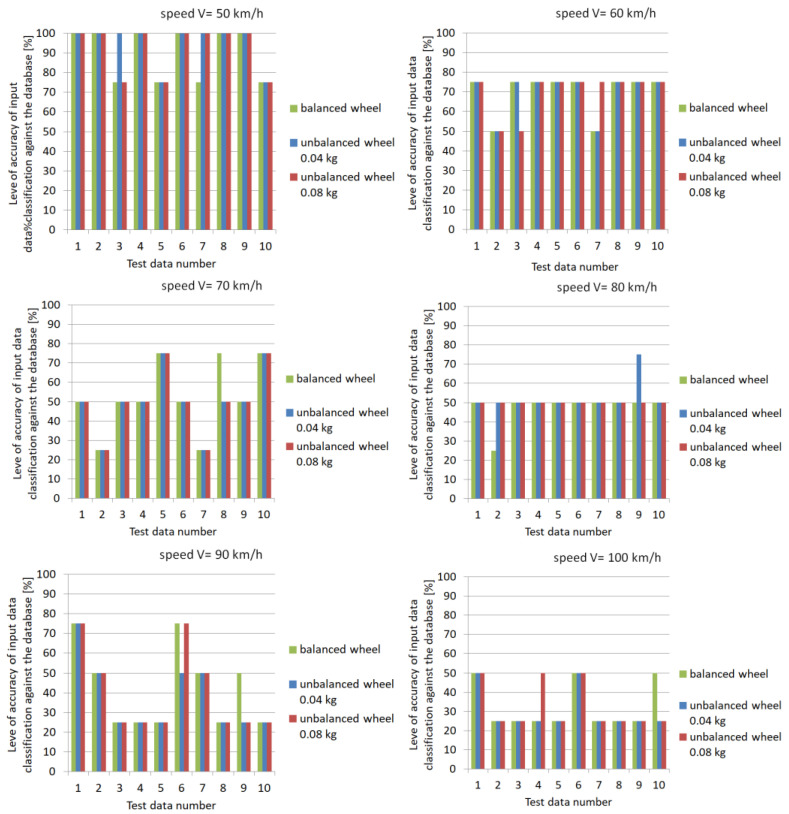
Level of accuracy of input data classification against the database.

**Table 1 sensors-23-02326-t001:** Acceleration sensor type 3DM-GX3-25 specifications.

Measurement range	±5 g
Non-linearity	±0.1% fs
In-run bias stability	±0.04 mg
Initial bias error	±0.002 g
Scale factor stability	±0.05%
Noise density	80 μg/√Hz
Data output rate	1000 Hz

**Table 2 sensors-23-02326-t002:** Datron L-350 Aqua specifications.

Speed range	0.3–250 km/h
Distance resolution	1.5 mm
Distance measurement deviation	<±0.1%
Speed linearity	<±0.2%
Working range linearity	<±0.2%

**Table 3 sensors-23-02326-t003:** An excerpt from the learning data table.

In	In	In	Out
*x*	*y*	*z*	Wheel Condition
0.117304	0.103775	0.039480	nw40d1
0.062202	0.052450	0.059505	nw40d2
0.022630	0.024011	0.051509	nw40d3
0.062550	0.044582	0.055876	nw40d2
0.049989	0.014870	0.035145	nw40d2
0.126599	0.085579	0.056270	nw40d6

## Data Availability

Not applicable.

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
