# Peer review of "Diagnosis of the Pneumatic Wheel Condition Based on Vibration Analysis of the Sprung Mass in the Vehicle Self-Diagnostics System"

_sensors, 2023, doi:10.3390/s23042326_

Round 1

Reviewer 1 Report

Details about the technical specification of the sensors used should be included. What was the sensitivity? What was the sampling frequency set while acquiring vibrations?

Was the data normalized/ standardized?

Hyperparameters of the decision tree must be included.

Which criterion was used while training the decision tree? Was it Gini or Entropy? Why?

Did you check the influence of ‘max_depth’ and ‘min_samples_split’ on testing the tree? If not, this study is a must.

Was the decision tree trained using standard hyperparameters, or were they altered?

How to deal with the data diversity of the present moment and moment in the future?

How to ensure the robustness of the model in a high noisy environment?

Include a discussion regarding the use of different methods for dynamic data distribution in the introduction. You may refer to some recent papers, Health Monitoring of Milling Tool Inserts Using CNN Architectures Trained by Vibration Spectrograms, Augmentation of Decision Tree Model Through Hyper-Parameters Tuning for Monitoring of Cutting Tool Faults Based on Vibration Signatures, A machine learning approach for vibration-based multipoint tool insert health prediction on vertical machining centre (VMC)

Comment on computational time and complexity in the training of the tree.

It is unclear how much data was used for the training and testing of trees. What was the split? Additionally, results can be provided considering different holdout % and holdout validation approaches. Refer to this article to understand the holdout validation approach. You may refer to the following article https://doi.org/10.1115/1.4051696

What are the limitations and future scope?

Check that the abstract provides an accurate synopsis of the paper. It is very vague in its present form.

Overall writing of the paper is in colloquial or oral language. The grammar is improvable. There must be a thorough proofreading of the paper.

Author Response

Dear Reviewer,

the authors of the article would like to thank the reviewer for comments and suggestions to the reviewed article. The comments provided will significantly improve the quality, transparency and scientific value of the article.

In order to ensure the transparency of the responses to the comments contained in the submitted review.  The reviewer's suggestions presented in the review were included in the article. Comments from other reviewers were also included in the revised publication. Detailed resposes are included in the file

Authors

Reviewer 2 Report

Dear Authors,

the manuscript entitled "Diagnosis of the pneumatic wheel condition based on vibration analysis of the sprung mass in the vehicle self-diagnostics system" by Krzysztof Prażnowski and co-workers discusses the use of method for identifying the occurrence of tire wheel unbalance based on vehicle body vibrations using an expert program. The paper is well justified, planned and written, and adds to the diagnostic methods knowledge some new informations. The introduction make sense with the topic and is crearle defined as well as the second chapter (Related works and other methods). In my opinion, the methodology and results of this study are clear. Conclusions are consistent with the findings. I appreciate the contribution that the Author made in preparing the manuscript. However, in my opinion the manuscript needs to be improved in some fields and some general remarks as well as the specific comments are bellow.

Evaluation of the paper, general remarks, editorial comments/typos:

1) Please add more References in the Introduction and Related works and other methods chapters. Example of publications that can be included in these chapters are:

- https://doi.org/10.4271/09-11-01-0006,

- http://dx.doi.org/10.1016/j.ymssp.2014.12.017.

2) In line 142, the Authors use the personal form ("… practical ways in which they are used...."). This is not correct in high-quality articles. It suggests modifying this part of the article. Please check the entire article in terms of personal form (line 227 and 381).

3) line 161 - there is [36], there is no such article in references. 

4) line 72, 289, 416 - the dot is missing at the end of the sentence.

5) line 447 - please translate the description in the Figure16 to English language.

6) Figure 22 - please modify the Y axis, there is 00% and should be 100%.

7) Authors present their results but without any discussion supported by the literature. When the results are not discussed and conveniently supported by the open literature, questionable conclusions are obtained. Currently, the article looks more like a report from test than a scientific article. Improvement in the description of the test results is required.

8) Research articles should present the directions of further research. I suggest adding one paragraph in the 7. Conclusion chapter.

Article is interesting but the modifications should be implemented before considering the manuscript for publication. I hope these suggestions can help to improve the quality of this paper.

I wish you all the best.

Author Response

Dear Reviewer,

the authors of the article would like to thank the reviewer for comments and suggestions to the reviewed article. The comments provided will significantly improve the quality, transparency and scientific value of the article.

In order to ensure the transparency of the responses to the comments contained in the submitted review.  The reviewer's suggestions presented in the review were included in the article. Comments from other reviewers were also included in the revised publication. Detailed resposes are included in the file.

Authors

Round 2

Reviewer 1 Report

The authors have addressed all my comments positively. The paper is well revised — my congratulations to the authors.